# Analysis of Lower Extremity Motor Capacity and Foot Plantar Pressure in Overweight and Obese Elderly Women

**DOI:** 10.3390/ijerph20043112

**Published:** 2023-02-10

**Authors:** Min Liu, Ning Kang, Dongmin Wang, Donghui Mei, Erya Wen, Junwei Qian, Gong Chen

**Affiliations:** 1Institute of Population Research, Peking University, Beijing 100871, China; 2Department of Physical Education, Peking University, Beijing 100871, China; 3College of Psychology, Capital Normal University, Beijing 100048, China

**Keywords:** lower extremity motor capacity, foot plantar pressure, overweight and obese older women, gait analysis

## Abstract

Background: Overweight, obesity and falls are major public health problems and old people are the biggest group suffering falls. Methods: 92 females were divided into the overweight or obesity (O) group (68.85 ± 3.85) and regular-weight (R) group (67.90 ± 4.02). Lower extremity motor capacity and plantar pressure were compared between the two groups. The IRB approval number is 20190804. Results: (1) Functional Movement Screen and Fugl-Meyer Assessment scores in the O group were significantly lower than in the R group. The time to complete the Timed Up and Go test in the O group was significantly longer than in the R group. (2) Foot flat phase, double support distance, and left foot axis angle in the O group were significantly higher than in the R group. Distance and velocity, left-foot minimum subtalar joint angle and right-foot maximum subtalar joint angle in the O group were significantly shorter than in the R group. (3) Peak force, average force and pressure of metatarsal 1–4, mid-foot, heel medial and lateral, peak pressure of metatarsal l, midfoot, heel medial and lateral in the O group were significantly higher than in the R group. (*p* < 0.05). Conclusions: Overweight and obese elderly women have a lower sensorimotor function, flexibility and stability in functional movements, but higher loads on the foot.

## 1. Introduction

The world’s population is rapidly aging. The number and proportion of people aged 60 and above are increasing. Today, the number of older persons worldwide is slightly more than 1 billion (13.5% of the global population), and 1 in 6 persons will be aged 60 years or older by 2030 [1]. By 2050, the global population of older people will increase to 2.1 billion (22.0% of the global population) [2] and China will have the largest population of older persons [3]. According to the *National Physical Fitness Monitor Bulletin* in 2022, the overweight and obesity rate among the elderly in China was 58.4% in 2020, an increase of 2.9% from 2014, and older women had a higher percentage of body fat than older men [4]. Falls are the second leading cause of unintentional injury deaths worldwide, and older persons suffer the most significant number of fatal falls [5]. Overweight older adults are at higher risk of falls due to their heavy weight, heavy lower extremity burden, lower stability and weaker flexibility [6]. As the major public health problems, overweight, obesity and falls have seriously affected the healthy life of the elderly in China, and also in many other countries.

A large number of studies have demonstrated that falls among older people result from diverse and complex causes [7,8,9,10]. Most falls occur in mobile activities such as walking, running, hiking or climbing, and changing directions [11,12,13] because of poor mobility [5]. The ability to be mobile is an essential condition of a high-quality and healthy life. Still, it is subordinated to functional capacity, which is critical to realizing healthy aging [14]. However, with the increase in age, the ability to be mobile declines [15]. The changes in physical structure and function, including instance, gait stability, muscle strength, joint flexibility, postural and neural control, etc., could cause functional limitation in the mobility of the elderly to varying degrees [16,17,18,19]. In addition, significant body weight could make the force on the joints increase, thus aggravating the load on the lower extremities of overweight and obese older adults in daily mobile activities, which would increase the risk of falling [20,21]. Therefore, it is of great significance to analyze the mobility of overweight and obese older adults to prevent falls, improve their quality of life and achieve healthy aging.

Walking is a mobile activity that older adults repeat most frequently in their daily lives. In the process of walking, the feet have the most direct contact with the ground. The pressure distribution between the plantar and the support surface during walking reflects the physiological, structural and functional information of the lower extremities and even the whole body [22]. In addition to feet, the completion of mobile activities is inseparable from the functional help of the lower extremities. Previous studies have indicated that the motor capacity measurement of lower extremities had closer associations with mobile activity performance and falls [23,24,25,26,27]. Similarly, it has also been shown that the distribution analyses of plantar pressure have great significance in studying the foot performance of physiological functions, fall prevention and bio-mechanical characteristics [28,29].

Therefore, in this study, under the background of healthy aging and fall prevention, we investigated the lower extremity motor capacity and the distribution of plantar pressure of overweight and obese older adults to explore the difference in lower extremity motor capacity and plantar pressure parameters between overweight and obese older women and regular-weight older women.

## 2. Methodology

### 2.1. Participants

In this study, we collected the data of older women who satisfied the following five criteria: (1) female; (2) age > 60 years old but <75 years old; (3) body mass index (BMI) > 18.5 kg/m^2^; (4) a score on the Lawton Instrumental Activities of Daily Living (IADL) Scale ≥ 6 [30], which means clear consciousness and independent walking without the use of AIDS; and (5) no lower extremity joint operation in the last two years, no prominent bone and joint disease, no heart disease, etc. In order to obtain enough participants, we recruited two community workers from three communities in the Haidian District of Beijing. These communities have well-constructed community information management systems. Thus, we determined the sample size based on the number of resident registrations. Up to the month before the formal test, the total number of old people between the age of 60 to 75 was 978. A total of 10% of the number of old people were selected for the study. The sample size was then expanded by 15% to take into account the dropout rate, which resulted in a sample size of 978 × 10% × (1 + 15%) = 113. Finally, 115 older people (110 females and 5 males) were recruited for the survey. However, the number of males was too small. Thus, in order to eliminate of gender influence and ensure a sufficient sample size, five males were excluded and only 110 females were included. All subjects were required to undergo a series of tests. After excluding 12 subjects who did not meet the criteria, we selected 98 older females for the study. Based on the BMI classification standard from the World Health Organization [31], the elderly with BMI ≥ 25 kg/m^2^ were divided into the overweight or obesity group (the O group), and the elderly with 18.5 < BMI < 25 kg/m^2^ were divided into the regular-weight group (the R group). One person in the regular-weight group and five persons in the overweight or obesity group were excluded when doing the data analysis because they didn’t complete all of the tests. Finally, 46 and 46 people were enrolled in the two groups. The sample size in the study was 10.02%. More detailed information is provided in Figure 1.

### 2.2. Experiment Time and Place

All tests were carried out in the Innovation Laboratory of the Integration of Sport and Medicine at Peking University from 25 September 2022 to 12 October 2022. The average temperature in the laboratory was 21.2 °C, and the average humidity was 38.5%, which conformed to the machine requirements (operating temperature range 15 °C to 30 °C and humidity range of 20% to 80%) to maintain the optimal performance of the instruments, especially the Footscan system. If the Footscan system was operated outside the specified environmental temperature and humidity parameters, the accuracy of the data collection would be affected.

### 2.3. Instrument and Measures

The following outcomes were measured for all participants.

#### 2.3.1. Measure of Participant Characteristics at Baseline

Body composition outcomes were performed by the Inbody 270 produced by Korea Inbody Co. Ltd. (Seoul, Republic of Korea). The machine conducted normalized, self-service, national body composition outcome testing, assessing height (m), weight (kg), body mass index (kg/m^2^) and body fat (%).

The instrumental activity of daily living (IADL) was tested by the Lawton Instrumental Activities of Daily Living Scale (Lawton-IADL) [32]. Lawton-IADL is a highly reliable and validated instrument to assess a person’s ability to perform independent living skills [30,33]. It measures eight domains, including using a telephone, shopping, preparing food, housekeeping, laundry, transportation, medications, and finances [30]. The summary score ranges from 0 (low function) to 8 (high function), with higher scores indicating better independent living ability.

Global risk analysis was shown in the risk analysis screen in the Footscan 9^®^ software (version 9.5.7), which is produced by the Belgium Footscan Company (Olen, Belgium). The global risk of feet is a predictive percentage value calculated based on the Footscan risk analysis algorithm [34] by using the D3D design wizards with a dynamic measurement method. The global risk analysis could provide a possible percentage to evaluate the injury risk for the left and right foot together. The possible global foot risks are divided into low, medium and high levels. This was scientifically validated in a prospective cohort study [34], which proved the predictive value in the correct prediction of lower limb injury risk.

In addition to the above characteristics, all subjects’ age (years) and length of foot (EU) were also collected.

#### 2.3.2. Measure of Lower Extremity Motor Capacity

The Functional Movement Screen (FMS), proposed by Cook and Burton [35], was used to assess the flexibility and stability in basic functional movements and to identify functional limitations and asymmetries of older active adults [36,37]. The full assessment of FMS contains seven different functional movements, including trunk or core strength and stability; neuro-muscular coordination; symmetry of movement; flexibility; acceleration; deceleration; and dynamic stability. Every assessment score ranges from 0 to 3, and the total score is 21, with higher scores indicating better functional movement.

The Fugl-Meyer Assessment (FMA), developed by Fugl-Meyer and colleagues, is regarded as a highly comprehensive scoring system to describe sensorimotor function [38]. Although it is a stroke-specific measurement [39], it can also reflect the functionality of lower extremities for other groups of older adults [40,41,42], even in the exploration of the prevention of falls [43]. It is divided into two assessment subscales in the upper and lower extremities. Both cover five domains: motor functioning, balance, sensation, joint motion, and joint pain [44]. The lower-extremity motor subscale of the Fugl-Meyer Assessment (FMA-LE) comprises seven categories with 17 tasks. The seven categories are reflex activity, hyperreflexia, flexor synergy, extensor synergy, movement combining synergies, movement out of synergy and the heel-knee-shin test. Each task is scored on a scale from 0 to 2, with a score of 0 indicating that the subject is unable to perform the task, a score of 1 indicating that the subject can partially perform the task, and a score of 2 indicating that the subject can fully perform the task. The total score is 34, with higher scores indicating better sensorimotor function.

The timed Up and Go (TUG) test, developed by Podisadle and Richardson [45], is a rapid assessment of functional walking ability with high sensitivity and specificity [46]. The test only requires one chair (0.45 m tall) and a cone. The cone is placed 3 m away from the chair. The test records the time taken for a subject from leaving the chair surface, walking as fast as possible, going around the cone, to return to sit on the chair (hip touching the chair). The subjects had the chance to practice once and were then tested twice at the one-minute interval. The best result of the two tests was recorded in a shorter time (s), indicating better functional walking ability.

#### 2.3.3. Measure of Foot Plantar Pressure

The tests of foot plantar pressure were performed by the Footscan^®^ pressure measurement system produced by the Belgium Footscan Company. The system consists of a 1.5-m entry-level plate (dimensions: 1.61 m × 0.47 m × 0.018 m, weight: 24 kg), and the Footscan^®^ software (version 9.5.7) was developed to perform state-of-the-art foot plantar pressure recording and analysis.

The 1.5 m entry-level plate measures plantar pressure using an X-Y matrix of resistive pressure-sensitive sensors (data acquisition frequency: 200 Hz, number of sensors: 12,288, sensor dimensions: 0.00762 m × 0.00508 m, active sensor area: 1.46 m × 0.33 m, pressure range: 10,000–1,270,000 Pa) that are scanned sequentially.

The Footscan^®^ 9 software (version 9.5.7) accurately registers and records the dynamic plantar pressure data of subjects in barefoot as well as shod condition when the subject walks over the plate to conduct the plantar pressure distribution analysis of the complete gait cycle. More specifically, the 1.5 m entry-level plate measures plantar pressure and force using an X-Y matrix of resistive pressure sensitive sensors that are scanned sequentially. In addition, the dynamic measurement captures the pressure and force distribution under the subject’s feet over the full duration of a step from initial contact until the end of the foot roll-off. The value registered by the footscan device are proportionate to the pressure and force on each sensor up to an accurate measurement value. Its reliability and repeatability were well validated in previous studies [47,48].

Spatial-temporal gait parameters, foot axis parameters, force and pressure parameters, etc., could be recorded and analyzed by the system, where spatial-temporal gait parameters are based on different segmentation of foot force phases, and force and pressure parameters are based on different plantar pressure zones, while foot axis parameters are based on the foot axis line.

Segmentation of foot force phase: While walking, the heel strike to heel strike on the same side is one gait cycle. The activity of a gait cycle can be divided into the support time phase and the swing time phase. According to some key moments of initial foot contact (IFC), initial metatarsal contact (IMC), initial forefoot flat contact (IFFC), heel off (HO), and last foot contact (LFC), the single foot timing is divided into the initial contact phase (ICP), forefoot contact phase (FFCP), foot flat phase (FFP), and forefoot push-off phase (FFPOP). See Figure 2. Plantar pressure zones: the plantar pressure is divided into ten main zones based on anatomy, including the heel, middle foot, five metatarsal bones, and five toes. All of the partitions are as follows: toe 1 (T1), toe 2 to toe 5 (T2–5), metatarsal 1 (M1), metatarsal 2 (M2) and metatarsal 3 (M3), metatarsal 4 (M4), metatarsal 5 (M5), midfoot (MF), heel medial (HM), and heel lateral (HL) (see Figure 3). The foot axis line is the line connecting the middle of the MH and LH with the middle of M2 and M3 (see Figure 4). The long pink line shows the foot axis direction, the broken blue line shows the walk direction, and the brown lines express the minimum and maximum subtalar joint angles.


**Observed indicators:**


Spatial-temporal gait parameters: (1) distance (m), duration (s), velocity (m/s), stance duration (s), and swing duration (s) in one gait cycle; (2) heel to heel base of support distance of support middle foot (m), double support distance of support middle foot (m), single support distance of support middle foot (m); (3) single foot timing of about four phases for both feet.

Foot axis parameters: foot axis angle, maximum subtalar joint angle, minimum subtalar joint angle, and subtalar joint flexibility of both feet. (1) The foot axis angle is the angle between the foot axis direction and the walk direction, referring to the position of foot rotation related to the gait direction. A positive or negative value indicates external or internal rotation, respectively, of the foot in the horizontal plane [49]. (2) the minimum and maximum values, respectively, suggest the rearfoot’s maximal supination and maximal pronation position.

Force and pressure parameters: average force (N), average pressure (Pa), peak force (N) and peak pressure (Pa) in 10 plantar pressure zones of both feet.


**Test requirements:**


Place the plate flat on the floor and connect it to the computer. To achieve a stable state of walking and reduce test errors, a person is required to carry out free walking three times before the formal test to familiarize themselves with the walking environment. All subjects should take off their shoes and socks and then step gradually from the front of the plate to the test plate. During the whole walking process, the person should keep their eyes straight ahead, walk naturally with a moderate pace, and swing their upper extremities naturally. All subjects are required to walk over the plate 6–8 times.

All measurements were performed following the order of body composition test, FMS, FMA, TUG, and foot plantar pressure test. The timing of the every test would be controlled within 5 min and the total duration of a whole test is no more than 30 min.

### 2.4. Statistical Analysis

Microsoft Office 2020 and IBM SPSS Statistics (version 25; American IBM) were used for the three-part statistical analysis. Firstly, the Kolmogorov-Smirnov test (K-S test) [50] was used as the judgment method to test the normal distribution of the data.

Secondly, descriptive statistical methods were applied to present the older women’s basic characteristics in the R group and O group. If the data were normally distributed, the data were described as the means and standard deviations (M ± SD); if not, the data were described as medium and interquartile range, that is, Md (P_25_, P_75_).

Finally, if the data were normally distributed, the independent sample T-test was conducted for inter-group comparison; if not, the rank sum test of two independent samples of non-parametric analysis was conducted, and the method of a Mann-Whitney U test was adopted. In order to eliminate the errors caused by the individual differences of each subject, before inter-group comparison, the data were standardized by the extremization method to reflect the differences of indicators more accurately. The formula is as follows: Z_χ*i*_ = (χ*_i_* − Min*_i_*)/(Max*_i_* − Min*_i_*) × 100, (*i* = 1, 2, …, *p*)
where Z_χ*i*_ is the standardized value; χ*_i_* is the original value; Max*_i_* is the maximum of the original value, and Min*_i_* is the minimum of the original value.

The confidence interval (CI) was 95%, and *p* < 0.05 was considered a significant difference.

## 3. Results

### 3.1. Descriptive Statistics

A total of 92 females completed the tests, with 46 in the R group and 46 in the O group (36 overweight individuals and 10 obese individuals). Table 1 illustrates the basic description and the comparison results between the two groups. Except for weight, BMI and body fat, which were related to the basis of grouping, other characteristics of the two groups were not significantly different (*p* > 0.05) (Table 1).

### 3.2. Lower Extremity Motor Capacity

Table 2 shows the comparisons of lower extremity motor capacity comparison assessment results between the R group and the O group. The scores of FMS and FMA-LE in the O group were significantly lower than those in the R group (*p* < 0.05, *p* < 0.01), and the time of TUS in the O group was significantly longer than it was in the R group (*p* < 0.05).

### 3.3. Foot Plantar Pressure

#### 3.3.1. Spatial-Temporal Gait Parameters

Table 3 demonstrates the comparisons of spatial-temporal gait parameter assessment results between the R group and O group. In the aspect of the gait cycle, the distance and velocity of one gait cycle in the O group were significantly lower than those in the R group (*p* < 0.01, *p* < 0.05). In the aspect of the support middle foot, the double support distance in the O group was significantly higher than in the R group (*p* < 0.05). In the aspect of the single-foot timing, the foot flat phase in the left foot in the O group was significantly longer than in the R group (*p* < 0.05).

#### 3.3.2. Foot Axis Parameters

Table 4 shows the comparisons of foot axis parameter assessment results between the R group and the O group. The left foot axis angle in the O group was significantly larger than in the R group (*p* < 0.05). In contrast, the left minimum subtalar joint angle and right maximum subtalar joint angle were significantly smaller than those in the R group (*p* < 0.05, *p* < 0.01).

#### 3.3.3. Average Force and Pressure Parameters

Table 5 and Table 6 displayed the comparisons of average force and pressure assessment results in the different zones between the R group and O group. In summary, the total average forces in both feet in the O group were significantly higher than in the R group (*p* < 0.01). Among the ten plantar pressure zones, except for the average forces of T1–5, the average forces of the other eight zones in both feet in the O group were significantly higher than those in the R group (*p* < 0.05); except for the average pressures of T1–5 in both feet and M5 in the right foot, the average pressures of other zones in both feet in the O group were significantly higher than those in the R group (*p* < 0.05).

#### 3.3.4. Peak Force and Pressure Parameters

Table 7 and Table 8, display the comparisons of peak force and pressure assessment results in the different zones between the N and O groups, respectively. In summary, the total peak forces in both feet in the O group were significantly higher than in the R group (*p* < 0.01). Among the ten plantar pressure zones, except for the average forces of T1–5, the average forces of the other eight zones in both feet in the O group were significantly higher than those in the R group (*p* < 0.05); the average pressures of M3–5 in the right foot and M1, MF, HL, and HM in both feet in the O group were significantly higher than those in the R group (*p* < 0.05).

## 4. Discussion

Our main findings are as follows: (1) Compared with regular-weight older adults, overweight and obese older women had lower FMS, lower FMA-LE and higher TUG, which indicated that the lower extremity motor capacities were relatively poorer. (2) Compared with regular-weight older adults, overweight and obese older women had shorter distance and velocity of one gait cycle, as well as longer FFP and DSD, which showed that they had slow speed, short stride, great contract and long support time when walking. (3) Higher left foot axis angle, lower left minimum subtalar joint angle, and right maximum subtalar joint angle were observed in overweight and obese older women compared to non-obese older women. The results revealed the lack of stability in the rearfoot of overweight and obese older women. (4) Compared to non-obese older women, overweight and obese older women undertook more plantar forces and pressure in M1–5, MF, HL, and HM, particularly midfoot and rearfoot, which suffered more peak pressures.

In terms of lower extremity motor capacity, our study selected FMS [37], TUG [51] and FMA-LE [52] as the assessment measurement of functional mobility. It was shown that the significantly lower FMS, lower FMA-LE, and higher TUG scores of overweight and obese older women indicated that lower extremity motor capacity is adversely affected by excess body weight. This finding of FMS and FMA-LE are consistent with the studies which all reported the negative pairwise correlations between the FMS and BMI [37,53] and FMA-LE and BMI [38,54]. Previous studies also revealed similar results for TUG [55,56], in that the total time of TUG was longer in obese women than in regular-weight women. Herein, we confirmed that the high weight may limit neuromuscular control, mobility, and stability [37,57]. Those limitations in these areas can result in poor performance in lower extremity motor capacity and eventually lead to a high risk of falls in overweight and obese older adults.

In the aspects of spatial-temporal gait parameters, previous studies on obese adults have found that individuals with high BMI had a lower absolute walking velocity, a smaller absolute stride length, and a longer stance duration when walking freely [58,59]. They also found that when the spatial-temporal gait parameters were stratified by age, velocity and distance of one cycle, there was a pronounced decline with increasing age [60]. The findings of our study not only concur with the above findings but also showed a longer double support distance and foot flat phase when walking, similar to the literature, which also reported a longer contact duration [61] and a longer double support time [59]. The previous study revealed a close association between a slower gait speed and a higher risk of falls in old adults [62]. If we try to explore the reason for this, it is probable that the long-term big loads to the lower extremities in heavier individuals may bring about more pain and discomfort, so as to result in lower walk speed and distance, as well as longer foot flat phase duration and the double support distance of the support middle foot to maintain a steady walk.

Regarding foot axis parameters, foot axis angle, subtalar joint angle and flexibility were recorded. The value of the foot axis angle provides the degree of internal or external rotation of the foot related to the gait direction. Our study found that overweight and obese older women had larger positive foot axis angles than regular-weight older women, which suggests a larger external rotation of the foot related to the gait direction. This finding is in line with those which are typically displayed in overweight and obese children during the gait [49,63]. The reason why heavier persons have larger foot axis angles may be because they need to maintain stability by widening the angle of the foot rotation and the contact area with the ground.

Nevertheless, toe-out foot was easier for people with larger external rotations, which may imply a more severe problem with tibial rotation [49,64]. Thus, heavier individuals find it more challenging to keep their balance in long-distance walking. The subtalar joint is a mitered hinge to translate the transverse plane motion of the tibia into the frontal plane motion of the foot [65]. The subtalar joint angle provides the amount of frontal plane rearfoot motion in relation to the ground during the initial contact phase, where the minimum and maximum values suggest the maximal supination and maximal pronation position of the rearfoot, respectively. A higher value for the subtalar joint angle indicates a more pronated rearfoot. Subtalar joint movement is classified as inversion-eversion [66], but due to the differences in techniques and devices, the subtalar joint movement is reported to have a high variability in different studies [67,68,69]. In addition, affected by several adjacent joints, ligaments and periarticular tendons, subtalar joint anatomy and movement are complex, and subtalar joints are highly variable between individuals [66]. The previous study has pointed out that body weight is a key parameter correlated to the supination of the rearfoot [70]. Our study observed a lower left-foot minimum subtalar joint angle and right-foot maximum subtalar joint angle in heavier older women. The larger maximum subtalar joint angle is probably related to the reason the more pronated rearfoot has some difficulties in supporting the arch in time during the contact phase, which easily leads to injury. In contrast, the larger absolute value of the minimum subtalar joint angle may be the reason that the rearfoot sees less pronation and more supination is locked. The connective tissue elasticity cannot be fully utilized to absorb the impulse, so these instantaneous loads are directly imposed on the joint, increasing the probability of joint injury and muscle fatigue. These findings revealed weaker walking stability in the rearfoot and more variations of the subtalar joint morphology in overweight and obese older women. However, there are few studies on the foot axis angle and subtalar joint angles of overweight and obese older adults, which requires further investigation.

As for the force and pressure parameters, our study compared overweight and obese older women with non-obese older women and found that heavier older women undertook more plantar forces and pressure, particularly on the metatarsal, midfoot and heel. The findings of our study are in line with the recent findings [71,72,73], which also exhibited similar results indicating that high BMI and weight can contribute to high plantar force and pressure. Excessive plantar pressure and force are significantly related to foot pain [74]. As for the reason for this, it would appear that the excess fat tissue can generate greater overload to give rise to excessive stress/strain on the musculoskeletal structures of the foot [59]. Additionally, a long-term overload accumulation also can result in the collapse of the medial longitudinal arch, causing a greater contact area of the foot with the ground [75], which is another cause of the increase in foot pain.

To explore whether excessive body mass could affect the results of this study, a comparison between overweight (25 ≤ BMI < 30 kg/m^2^) and obese (BMI ≥ 30 kg/m^2^) [31] old people was performed. It was shown that except for the peak force of M 1 and 2 in the right foot, other indicators had no significant difference, which indicated that the excessive weight has little influence on our results, but if the BMI fell outside the normal criteria, the lower extremity motor capacity and foot plantar pressure would obviously be different. Therefore, we strongly suggest that overweight and obese old individuals should reduce excess fat mass to maintain a regular weight. In addition, we also suggest that old people should wear comfortable shoes and adjust their walking posture to relieve the loading of foot structures and reduce foot pain and discomfort. In addition, they should strengthen their balance, strength and coordination exercises to enhance lower extremity stability and control of the foot when walking, reducing the risk of falls and contributing to the maintenance of a high quality of life.

The significance of our study lies in that we discovered the characteristics of lower extremity motor capacity and plantar pressure distributions in overweight and obese older women by comparing them with regular-weight older women. Although we did not determine the correlation between these characteristics and body weight, we can further evaluate the correlations in the future by recruiting more subjects, setting up more groups and using more objective analytical methods. Additionally, more fall-related indicators need to be collected. In that way, we will be able to obtain more detailed findings between body weight and those characteristics.

Several limitations are present in this study. The findings of our study are based on a small sample of fewer than 100 people and only two groups. Due to the small sample, we could not classify subjects into more detailed groups according to BMI. Therefore, it is possible that the results would have been different when involving more groups of subjects. In following studies, we will continue to recruit more subjects and expand the sample size and groups, hoping to observe more scientific and objective indicators to provide more value for our healthy aging research.

## 5. Conclusions

In conclusion, our study shows that overweight and obese older women have the characteristics of poorer lower extremity motor capacities, a shorter distance and velocity of gait cycle, and a greater contract phase and longer support times when walking. In addition, they have higher foot axis angles, lower left minimum subtalar joint angles, right maximum subtalar joint angles and greater plantar pressures. Therefore, older adults with overweight and obesity suffer weaker stability, limited motor function, more foot pain and discomfort, and even a higher risk of falls.

## Figures and Tables

**Figure 1 ijerph-20-03112-f001:**
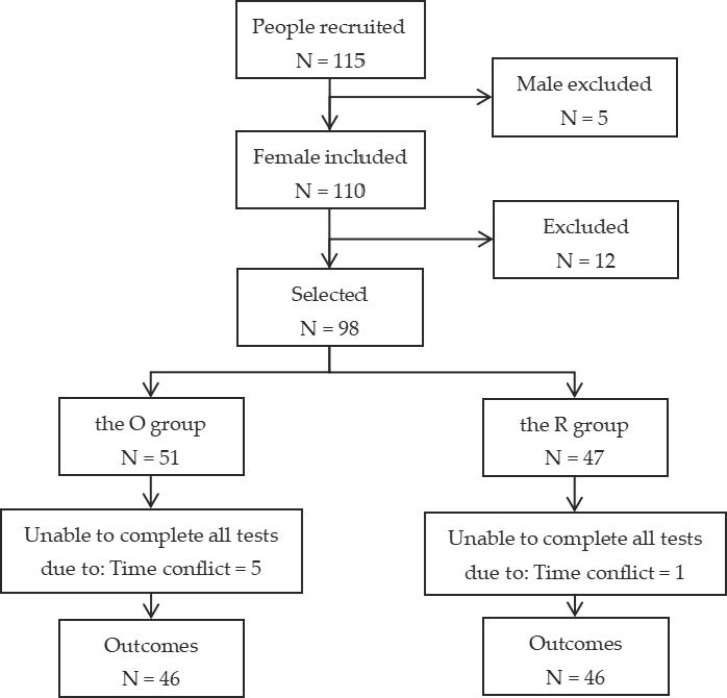
Flowchart explaining the assignment of the participants to O and the R groups.

**Figure 2 ijerph-20-03112-f002:**
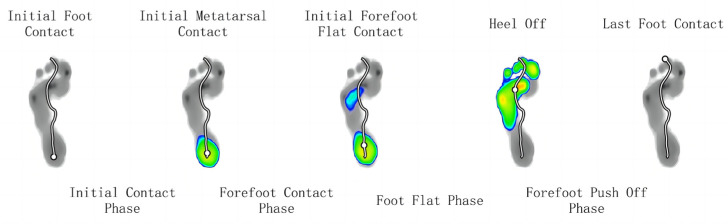
Segmentation of single foot timing.

**Figure 3 ijerph-20-03112-f003:**
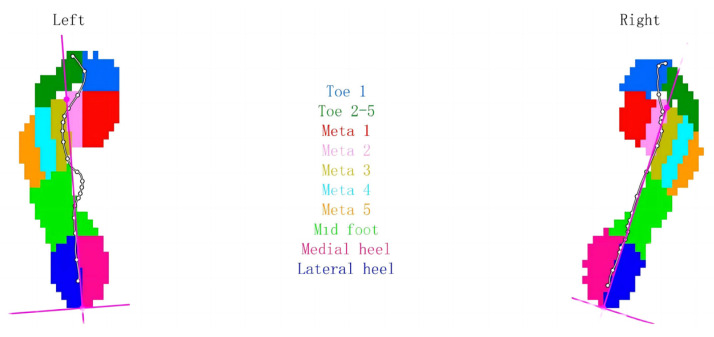
Plantar pressure zone diagram.

**Figure 4 ijerph-20-03112-f004:**
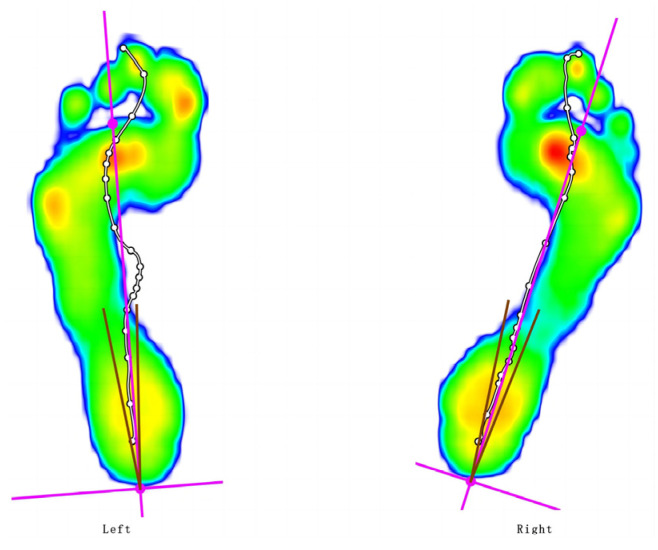
Foot axis angle.

**Table 1 ijerph-20-03112-t001:** Characteristics of the older women in O and R groups.

	Overall	The R Group	The O Group	Statistics (t/z)	*p* Value
	(*n* = 92)	(*n* = 46)	(*n* = 46)		
Age (Years)	68.38 ± 3.94	67.90 ± 4.02	68.85 ± 3.85	−1.165	0.247
Height (m)	1.57 ± 0.06	1.57 ± 0.05	1.57.13 ± 0.07	0.575	0.567
Weight (kg)	62.60 ± 11.23	54.63 ± 6.05	70.57 ± 9.40	−7.046	0.000 **
BMI (kg/m^2^)	25.21 ± 4.14	21.90 ± 1.86	28.52 ± 2.95	−8.262	0.000 **
Body Fat (%)	38.03 ± 7.20	32.61 ± 5.12	41.54 ± 6.12	−4.715	0.000 **
IADL	8.00 (7.00, 8.00)	8.00 (7.00, 8.00)	8.00 (7.25, 8.00)	−0.961	0.341
Global risk	1.00 (1.00, 1.00)	1.00 (1.00, 1.50)	1.00 (1.00, 1.00)	−0.768	0.445
Shoe Size (EU)	38.09 ± 1.56	37.92 ± 1.52	38.26 ± 1.60	−1.036	0.303

Notes: (a) Normally distributed values are expressed as mean ± standard deviation (M ± SD); not normally distributed values are expressed as Md (P_25_, P_75_). (b) BMI refers to body mass index. (c) Inference: ** *p* < 0.01. (d) Except for weight, BMI and body fat, there were no significant differences among other characteristics between the two groups (*p* > 0.05).

**Table 2 ijerph-20-03112-t002:** Lower extremity motor capacity of the older women in the O and R groups.

	Overall	The R Group	The O Group	Statistics (t/z)	*p* Value
	(*n* = 92)	(*n* = 46)	(*n* = 46)		
FMS	16.50 (14.46, 18.75)	18.00 (15.50, 19.00)	15.00 (11.04, 18.38)	−2.601	0.009 **
FMA-LE	32.00 (28.00, 34.00)	33.00 (31.00, 34.00)	30.00 (27.00, 34.00)	2.092	0.041 *
TUG (s)	6.56 ± 1.29	6.12 ± 0.83	6.87 ± 1.47	−2.021	0.043 *

Notes: (a) FMS refers to Functional Movement Screen, FMA-LE refers to the lower-extremity motor subscale of Fugl-Meyer Assessment, and TUG refers to Timed Up and Go. (b) Inference: ** *p* < 0.01, * *p* < 0.05.

**Table 3 ijerph-20-03112-t003:** Spatial-temporal gait parameters of the older women in the O and R groups.

	Overall	The R Group	The O Group	Statistics (t/z)	*p* Value
	(*n* = 92)	(*n* =46)	(*n* =46)		
Gait cycle					
Distance (m)	1.02 (0.86, 1.08)	1.06 (0.95, 1.08)	0.88 (0.82, 1.01)	−3.143	0.002 **
Duration (s)	1.12 ± 0.15	1.11 ± 0.13	1.13 ± 0.16	−0.672	0.504
Velocity (m/s)	0.89 ± 0.18	0.94 ± 0.17	0.84 ± 0.17	2.611	0.011 *
Stance duration (s)	0.76 ± 0.11	0.74 ± 0.10	0.77 ± 0.12	−1.464	0.147
Swing duration (s)	0.36 (0.33, 0.39)	0.37 (0.33, 0.44)	0.35 (0.33, 0.38)	−1.918	0.055
Support middle foot					
HHBSD (m)	0.06 ± 0.03	0.05 ± 0.02	0.06 ± 0.03	−1.212	0.229
DSD (m)	3.66 × 10^−4^ ± 4.82 × 10^−5^	3.44 × 10^−4^ ± 8.17 × 10^−5^	3.82 × 10^−4^ ± 7.56 × 10^−5^	−2.159	0.034 *
SSD (m)	3.63 × 10^−4^ ± 8.05 × 10^−5^	3.73 × 10^−4^ ± 4.13 × 10^−5^	3.59 × 10^−4^ ± 5.37 × 10^−5^	1.265	0.210
Single foot timing					
Left foot					
ICP (s)	30.00 (20.00, 45.00)	25.00 (20.00, 40.00)	20.00 (10.00, 28.75)	−1.921	0.055
FFCP (s)	25.00 (15.00, 40.00)	20.00 (15.00, 50.00)	35.00 (20.00, 40.00)	−1.147	0.251
FFP (s)	419.21 ± 148.09	378.64 ± 125.36	458.89 ± 158.89	−2.641	0.010 *
FFPOP (s)	255.00 (207.50, 282.50)	260.00 (227.50, 297.50)	242.50 (190.00, 307.50)	−1.314	0.189
Right foot					
ICP (s)	30.00 (20.00, 40.00)	35.00 (12.50, 45.00)	25.00 (15.00, 35.00)	−1.126	0.260
FFCP (s)	27.08 ± 19.98	24.55 ± 18.83	29.56 ± 20.97	−1.185	0.239
FFP (s)	424.51 ± 114.33	408.18 ± 94.53	440.47 ± 129.94	−1.343	0.183
FFPOP (s)	247.81 ± 55.18	245.57 ± 41.22	250.00 ± 66.48	−0.379	0.706

Notes: (a) HHBSD refers to the heel-heel base of support distance. DSD refers to double support distance. SSD refers to a single support distance. ICP refers to the initial contact phase, FFCP refers to the forefoot contact phase, FFP refers to the foot flat phase, and FFPOP refers to the forefoot push-off phase (b) Inference: ** *p* < 0.01, * *p* < 0.05.

**Table 4 ijerph-20-03112-t004:** Foot axis parameters of the older women in the O and R groups.

	Overall	The R Group	The O Group	Statistics (t/z)	*p* Value
	(*n* = 92)	(*n* =46)	(*n* =46)		
Left foot					
Foot axis angle (°)	10.60 (7.15, 14.90)	9.10 (6.85, 19.60)	12.00 (7.60, 16.18)	−2.418	0.016 *
Maximum subtalar joint angle (°)	6.98 ± 5.52	7.09 ± 5.32	6.87 ± 5.77	0.190	0.849
Minimum subtalar joint angle (°)	−6.34 ± 4.43	−5.36 ± 3.33	−7.29 ± 5.15	2.089	0.040*
Subtalar joint flexibility (°)	12.00 (8.00, 16.00)	11.00 (8.00, 16.50)	11.50 (8.00, 20.25)	−0.679	0.497
Right foot					
Foot axis angle (°)	11.64 ± 5.39	11.51 ± 5.65	11.78 ± 5.17	−0.234	0.816
Maximum subtalar joint angle (°)	6.19 ± 5.70	7.93 ± 5.97	4.49 ± 4.92	2.972	0.004 **
Minimum subtalar joint angle (°)	−7.28 ± 3.95	−6.93 ± 4.05	−7.62 ± 3.86	0.823	0.413
Subtalar joint flexibility (°)	13.00 (8.00, 17.00)	14.00 (11.50, 20.50)	11.50 (8.00, 15.00)	−1.545	0.122

Inference: ** *p* < 0.01, * *p* < 0.05.

**Table 5 ijerph-20-03112-t005:** The average force of the older women in the O and R groups.

	Overall	The R Group	The O Group	Statistics (t/z)	*p* Value
	(*n* = 92)	(*n* =46)	(*n* =46)		
Left foot (N)					
T1	26.66 ± 12.30	25.98 ± 13.13	27.33 ± 11.52	−0.524	0.602
T2–5	12.62 ± 7.14	12.75 ± 7.66	12.48 ± 6.66	0.175	0.862
M1	69.85 ± 26.77	58.58 ± 17.89	81.12 ± 29.50	−4.431	0.000 **
M2	63.92 ± 17.94	57.15 ± 13.27	70.69 ± 19.53	−3.890	0.000 **
M3	58.32 ± 14.89	53.52 ± 12.98	63.11 ± 15.26	−3.246	0.002 **
M4	45.19 (36.43, 58.26)	45.12 (35.12, 57.18)	57.81 (44.85, 67.57)	−3.053	0.002 **
M5	28.73 (21.94, 36.34)	24.53 (19.04, 36.08)	36.29 (27.95, 45.03)	−2.764	0.006 **
MF	41.19 (23.60, 71.35)	25.01 (17.38, 51.12)	62.09 (39.52, 85.74)	−4.061	0.000 **
HL	86.61 ± 20.88	68.45 ± 14.96	82.69 ± 17.48	−4.199	0.000 **
HM	75.57 ± 17.69	78.67 ± 17.60	94.55 ± 21.05	−3.924	0.000 **
Sum	523.10 ± 95.48	466.45 ± 56.51	579.76 ± 93.17	−7.053	0.000 **
Right foot (N)					
T1	25.44 (19.43, 34.66)	21.42 (18.37, 29.15)	29.74 (19.43, 40.29)	−1.226	0.220
T2–5	13.30 ± 7.63	13.09 ± 7.03	13.50 ± 8.26	−0.259	0.796
M1	68.11 ± 24.62	61.49 ± 19.59	74.73 ± 27.44	−2.664	0.009 **
M2	62.80 ± 16.85	57.37 ± 14.53	68.23 ± 17.39	−3.250	0.002 **
M3	58.27 ± 17.19	54.14 ± 15.70	62.41 ± 17.78	−2.365	0.020 *
M4	50.12 ± 16.88	44.80 ± 13.23	55.44 ± 18.53	−3.167	0.002 **
M5	30.31 (23.37, 37.36)	28.25 (21.58, 32.20)	35.90 (29.92, 42.73)	−3.740	0.000 **
MF	54.88 ± 35.46	40.47 ± 26.93	69.29 ± 37.32	−4.247	0.000 **
HL	87.42 ± 21.07	71.14 ± 15.70	83.11 ± 20.09	−3.184	0.002 **
HM	77.12 ± 18.91	81.66 ± 17.45	93.17 ± 22.92	−2.710	0.008 **
Sum	531.53 ± 95.80	481.15 ± 63.16	581.90 ± 96.87	−5.910	0.000 **

Notes: (a) T1 refers to toe 1, T2–5 refers to toe 2 to toe 5, M1 refers to metatarsal 1, M2 refers to metatarsal 2, M3 refers to metatarsal 3, M4 refers to metatarsal 4, M5 refers to metatarsal 5, MF refers to midfoot, HM refers to heel medial, and HL refers to heel lateral. (b) Inference: ** *p* < 0.01, * *p* < 0.05.

**Table 6 ijerph-20-03112-t006:** The average pressure of the older women in the O and R groups.

	Overall	The R Group	The O Group	Statistics (t/z)	*p* Value
	(*n* = 92)	(*n* =46)	(*n* =46)		
Left foot (Pa)					
T1	26,880.91 ± 10,203.76	26,269.59 ± 10,987.89	27,492.22 ± 9436.59	−0.573	0.568
T2–5	12,536.55 ± 5042.78	12,499.11 ± 5467.12	12,573.98 ± 4640.27	−0.071	0.944
M1	45,816.39 (34921.53, 57360.95)	38,878.16 (31,975.72, 50,242.07)	48,480.99 (41,708.31, 67,450.99)	−3.373	0.001 **
M2	77,334.61 (65,714.08, 93,470.04)	76,043.73 (64,630.86, 90,641.50)	79,330.48 (68,545.92, 94,966.33)	−2.093	0.036 *
M3	67,460.87 ± 14,784.70	63,497.41 ± 12,335.08	71,424.32 ± 16,054.97	−2.655	0.009 **
M4	56,839.94 ± 17,759.21	51,956.65 ± 15,581.97	61,723.22 ± 18,607.44	−2.729	0.008 **
M5	41,721.92 ± 17,181.63	38,415.60 ± 13,515.79	45,028.24 ± 19,797.76	−1.871	0.065
MF	20,817.42 (13,922.49, 29,426.71)	16,169.70 (11,623.74, 22,274.82)	28,440.04 (18,572.86, 37,892.40)	−4.506	0.000 **
HL	55,378.87 ± 13,048.38	47,829.44 ± 9903.62	57,618.22 ± 11,694.76	−4.332	0.000 **
HM	52,723.83 ± 11,847.04	50,536.13 ± 11,058.98	60,221.61 ± 13,193.05	−3.816	0.000 **
Right foot (Pa)					
T1	26,955.02 (21,220.22, 32,320.62)	23,853.90 (20,666.57, 27,992.73)	32,217.23 (21,369.69, 36,716.73)	−1.351	0.177
T2–5	12,457.06 ± 5006.99	12,034.52 ± 4740.08	12,879.60 ± 5278.61	−0.808	0.421
M1	43,463.79 (35,196.56, 51,370.40)	42,136.86 (30,959.35, 50,377.56)	47,336.37 (42,101.50, 67,738.83)	−2.186	0.029 *
M2	78,871.09 ± 20,898.03	74,618.90 ± 17,464.46	83,123.28 ± 23,263.43	−1.983	0.049 *
M3	65,473.93 ± 16,513.66	61,360.97 ± 14,269.91	69,586.89 ± 17,698.78	−2.454	0.016 *
M4	56,464.07 ± 16,026.29	50,937.94 ± 12,599.46	61,990.20 ± 17,268.80	−3.507	0.001 **
M5	42,639.95 ± 15,765.51	37,166.58 ± 11,097.90	48,113.32 ± 17,838.60	−3.534	0.001 **
MF	24,005.79 ± 11,073.49	18,990.71 ± 8061.39	29,020.87 ± 11,470.11	−4.852	0.000 **
HL	55,401.66 ± 12,992.14	48,948.19 ± 8919.82	57,492.66 ± 13,531.21	−3.576	0.001 **
HM	53,220.42 ± 12,179.40	51,562.80 ± 10,354.89	59,240.52 ± 14,282.51	−2.952	0.004 **

Inference: ** *p* < 0.01, * *p* < 0.05.

**Table 7 ijerph-20-03112-t007:** Peak force of the older women in the O and R groups.

	Overall	The R Group	The O Group	Statistics (t/z)	*p* Value
	(*n* = 92)	(*n* =46)	(*n* =46)		
Left foot (N)					
T1	117.19 ± 43.97	115.72 ± 45.03	118.65 ± 43.33	−0.319	0.750
T2–5	59.12 ± 33.33	61.07 ± 36.99	57.18 ± 29.50	0.558	0.578
M1	243.82 ± 73.60	217.26 ± 51.16	270.38 ± 83.03	−3.694	0.000 **
M2	191.79 ± 39.28	180.00 ± 36.56	203.58 ± 38.72	−3.003	0.003 **
M3	163.99 ± 37.37	155.95 ± 37.57	172.04 ± 35.78	−2.103	0.038 *
M4	143.87 ± 52.12	130.57 ± 44.66	157.17 ± 56.01	−2.518	0.014 *
M5	108.01 (81.70, 125.21)	100.01 (67.47, 125.80)	114.66 (99.05, 139.94)	−2.319	0.020 *
MF	183.80 ± 90.49	150.18 ± 90.72	217.42 ± 77.59	−3.821	0.000 **
HL	286.39 ± 53.94	228.52 ± 43.99	265.64 ± 46.98	−3.912	0.000 **
HM	247.08 ± 48.95	265.58 ± 40.12	307.20 ± 58.21	−3.993	0.000 **
Sum	839.19 ± 151.12	748.07 ± 95.70	930.32 ± 141.60	−7.232	0.000 **
Right foot (N)					
T1	119.49 ± 47.71	116.19 ± 45.93	122.78 ± 49.71	−0.660	0.511
T2–5	62.09 ± 29.39	63.88 ± 27.14	60.30 ± 31.68	0.583	0.561
M1	238.43 ± 63.04	223.91 ± 53.63	252.94 ± 68.77	−2.258	0.026 *
M2	194.05 ± 45.96	180.59 ± 42.74	207.51 ± 45.55	−2.923	0.004 **
M3	162.45 ± 39.91	153.45 ± 38.44	171.44 ± 39.73	−2.206	0.030 *
M4	148.78 ± 47.41	133.12 ± 42.01	164.45 ± 47.75	−3.341	0.001 **
M5	106.35 (89.52, 129.49)	101.73 (89.87, 107.43)	126.15 (106.09, 144.45)	−3.561	0.000 **
MF	192.97 ± 92.90	151.36 ± 71.44	234.59 ± 93.85	−4.786	0.000 **
HL	284.49 ± 54.97	233.68 ± 40.19	272.91 ± 55.52	−3.883	0.000 **
HM	253.29 ± 52.08	267.81 ± 47.15	301.17 ± 57.61	−3.039	0.003 **
Sum	862.78 ± 148.16	775.26 ± 100.31	950.30 ± 136.63	−7.005	0.000 **

Inference: ** *p* < 0.01, * *p* < 0.05.

**Table 8 ijerph-20-03112-t008:** Peak pressure of the older women in the O and R groups.

	Overall	The R group	The O group	Statistics (t/z)	*p* Value
	(*n* = 92)	(*n* =46)	(*n* =46)		
Left foot (Pa)					
T1	118,710.87 ± 34,943.65	118,261.00 ± 36,923.20	119,161.00 ± 33,249.20	−0.123	0.902
T2–5	52,401.09 ± 23,827.44	54,374.00 ± 28,941.80	50,428.00 ± 17,393.30	0.793	0.430
M1	166,177.17 ± 48,926.88	152,457.00 ± 36,275.80	179,898.00 ± 56,036.20	−2.788	0.006 **
M2	241,317.39 ± 48,576.23	233,948.00 ± 43,584.80	248,687.00 ± 52,545.80	−1.464	0.147
M3	185,850.00 ± 35,500.21	181,174.00 ± 35,864.20	190,526.00 ± 34,894.10	−1.268	0.208
M4	166,730.43 ± 56,164.24	157,163.00 ± 53,813.60	176,298.00 ± 57,410.00	−1.649	0.103
M5	131,200.00 (104,250.00, 162,925.00)	135,900.00 (106,800.00, 168,200.00)	143,650.00 (108,275.00, 185,350.00)	−1.171	0.241
MF	79,522.83 ± 29,111.06	67,204.00 ± 26,695.50	91,841.00 ± 26,284.10	−4.460	0.000 **
HL	180,371.74 ± 35,509.03	158,222.00 ± 29,561.10	183,615.00 ± 30,003.40	−4.089	0.000 **
HM	170,918.48 ± 32,253.10	167,113.00 ± 27,701.00	193,630.00 ± 37,723.40	−3.843	0.000 **
Right foot (Pa)					
T1	117,527.17 ± 35,191.90	116,580.00 ± 35,028.40	118,474.00 ± 35,716.10	−0.257	0.798
T2–5	52,373.91 ± 18,070.68	51,443.00 ± 18,020.20	53,304.00 ± 18,271.80	−0.492	0.624
M1	157,268.48 ± 41,329.51	149,828.00 ± 40,785.30	164,709.00 ± 40,958.50	−2.132	0.033 *
M2	241,368.48 ± 58,910.88	230,450.00 ± 50,528.70	252,287.00 ± 64,971.00	−1.799	0.075
M3	181819.57 ± 38636.52	171663.00 ± 33530.20	191976.00 ± 41031.30	−2.600	0.011 *
M4	166195.65 ± 47851.23	148150.00 ± 41187.10	184241.00 ± 47625.90	−3.888	0.000 **
M5	137664.13 ± 44452.74	125187.00 ± 35372.20	150141.00 ± 49259.80	−2.791	0.006 **
MF	83011.96 ± 31334.50	68717.00 ± 24830.90	97307.00 ± 30841.50	−4.897	0.000 **
HL	176443.48 ± 31647.63	159289.00 ± 24560.80	183028.00 ± 36468.70	−3.662	0.000 **
HM	171158.70 ± 33142.30	165904.00 ± 24879.00	186983.00 ± 34341.70	−3.371	0.001 **

Inference: ** *p* < 0.01, * *p* < 0.05.

## Data Availability

The data presented in this research are available on request from the corresponding authors. Because of privacy concerns, not all data are available.

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
