# Peer review of "Analysis of Lower Extremity Motor Capacity and Foot Plantar Pressure in Overweight and Obese Elderly Women"

_ijerph, 2023, doi:10.3390/ijerph20043112_

Round 1

Reviewer 1 Report

The manuscript titled: "Analysis of Lower Extremity Motor Capacity and Foot Plantar Pressure in Overweight and Obese Elderly Women" describes a study that compared normal weight older individuals with overweight/obese counterparts regarding their characteristics of lower extremity motor capacity and the distribution of plantar pressure. The authors reported that compared to normal weight elderly women, overweight and obese older women have higher risk for falls due to impaired sensorimotor function, walking, stability, flexibility and higher loads on the feet. Nonetheless, there are several items that should be addressed before this manuscript can be considered for publication.

General concerns

The manuscript has multiple unclear sections that deeply impact the clarity of the information the authors are trying to convey and the rationalization for several aspects of the study. Furthermore, the authors frequently use inadequate language, such as using the term “body-shapes” to define a category of body composition outcomes. The methods section needs a thorough improvement in the organization of it sub-sections and the discussion in generally very superficial, with little reflection about the biomechanical or physiological mechanisms underlaying the obtained results and its implications.

Concerns by section

Abstract

- The abstract is generally well written. However, according to the journal’s guidelines, the abstract should be composed by background as the first section and not objective.

- The abstract should be around 200 words in length. This abstract is 400 words long. It should be greatly reduced.

- Timed Up-and-Go test does not provide a score, but a time to complete in seconds.

Introduction

- Sequenced numbered citations should as [1-5], instead of [1][2][3][4][5]

- Sentence in lines 59-60 is incomplete.

- Through the manuscript the authors mention that obese individuals have a lower extremity burden. It is unclear what the authors mean by that.

- There is no justification nor rationale for why the study only investigated women.

Methods

- How would the laboratory’s temperature and humidity, and relative temperature and humidity, affect the data collection?

- It is unclear how the global risk is estimated and what does it entail. Ultimately it will affect its interpretation.

- How accurate are the force estimations from pressure in this system? Has it been validated?

Results

- Variables should be presented with SI units.

- Table 4 has no units.

- Titles of all tables state that they only show data for overweight and obese older women, when in fact they express the data for the R and the O groups.

Discussion

- How many overweight and obese individuals were in the O group? Is it possible that one or another can be affecting the O group’s results and interpretation?

- This section is generally written as a corroboration of existing literature. Instead, this should be the space for a deep interpretation of the findings, supported on literature. What do these results mean? How can they be explained (mechanisms)? Why are we seeing this? What are the implications of these findings in functional mobility and fall risk?

Reviewer 2 Report

Thank you for the opportunity to review your manuscript, Analysis of Lower Extremity Motor Capacity and Foot Plantar Pressure in Overweight and Obese Elderly Women.

The article does not reflect the purpose of the study. The title should reflect the purpose of the study, which according to the authors, is to predict the risk of falling.

Abstract.

The objective should be reworded. At present, it is not clearly defined and drafted.

It is necessary to give the correct significant data, not the approximate ones.

Investigating the lower limb motor capacity and plantar pressure distribution of overweight and obese older adults will give us quantitative data associated with fall risk. But the current design needs to be adequate for predicting fall risk in this population.

The study's objective again needs reframed to focus on what is being done and not on interpretations of what is being sought.

The methodology should start with the study design, a description of the study population and sample, and then the other aspects.

No sample size calculation was performed.

Line 102-104- Specify that the exclusion was from the data analysis as it was after the intervention.

The images are of poor quality and should be improved.

In explaining the variables, the timing of the measurements and the approximate order and duration of the tests are missing.

The actual significance data should be given in the results, not approximate ones.

The discussion begins with a summary of the findings.

Lines 423, 428 - The suggestions should not appear in the conclusions but in the discussion.

Lines 398-407 - I think it does not add anything.

Lines 413-415 - It is not a limitation.

This study has more limitations, and they should reflect on them.

Round 2

Reviewer 1 Report

The authors have addressed all my previous comments adequately.

Reviewer 2 Report

The authors have responded to all the issues raised by making the necessary amendments.

The article has improved enough to recommend its publication.